# Body Image Concerns and Body Weight Overestimation Do Not Promote Healthy Behaviour: Evidence from Adolescents in Lithuania

**DOI:** 10.3390/ijerph16050864

**Published:** 2019-03-09

**Authors:** Rasa Jankauskiene, Migle Baceviciene

**Affiliations:** 1Institute of Sport Science and Innovation, Lithuanian Sports University, Sporto 6, LT-44221 Kaunas, Lithuania; rasa.jankauskiene@lsu.lt; 2Department of Health, Physical and Social Education, Lithuanian Sports University, Sporto 6, LT-44221 Kaunas, Lithuania

**Keywords:** body image, nutrition habits, physical activity, adolescents, disordered eating

## Abstract

The present study aimed to explore the associations between body image concerns, body weight evaluation, disordered eating, nutrition habits, self-esteem, and physical activity (PA) in a mixed sample of adolescents of both genders. **Methods.** A total sample of 579 adolescents (299, 51.6% were girls) participated in this study. The participants ranged in age from 14–16 years old (*M* = 15.0, SD = 0.4). Respondents provided their answers filling in the questionnaires consisting of a battery of self-report questionnaires. An analysis of covariance was employed to test the hypothesis about the differences in body image concerns in the groups of BMI and body weight estimation controlling for gender. **Results.** Adolescents with a higher body mass index (BMI) and those overestimating body weight reported a higher body dissatisfaction (BD), a drive for thinness (DT), social physique anxiety (SPA), disordered eating, and lower self-esteem, but there were no differences in PA. Body weight overestimation was more prevalent in girls, yet body weight underestimation was more prevalent in boys. In girls, a higher BMI and body weight overestimation were associated with having less sweets, a lower frequency of having breakfast and for just a lower BMI alone—with consuming less fats, spreads, and oils. In boys, the BMI was associated with consuming less fruits and berries. The boys’ body weight overestimation was related to a lower reported number of meals, a lower frequency of breakfast, and a lower consumption of milk, cheese, yogurt, fats, spreads, and oils. Body dissatisfaction in boys and girls was negatively related to the number of meals, DT is related to a lower breakfast consumption (in girls), SPA was related to a lower nutrition score (in boys), the number of meals (boys and girls), and a lower use of meat and vegetables (boys). **Conclusions.** Adolescents with a higher BMI and body weight overestimation demonstrated higher body image concerns, lower self-esteem and a poorer eating-related behavioural profile. Body image concerns and body weight overestimation did not promote healthy behaviour in adolescents. It is critical to promote a positive body image, adequate body weight evaluation, self-esteem, and a healthy lifestyle in health promotion and health education programs for adolescents of both genders and different BMIs.

## 1. Introduction

Adolescence is a time of major body transformations due to a rapid pace of growth. This period is closely intertwined with nutrition and is influenced by many factors such as body image concerns and body dissatisfaction [1]. A plethora of studies have shown that adolescents’ body dissatisfaction is associated with a higher prevalence of unhealthy weight loss strategies, lower self-esteem, poorer perceived health, a longer time on the computer, an increased risk of eating disorders, suicidal ideation, lower physical activity, and dysfunctional exercise [2,3,4,5,6,7,8,9,10]. Furthermore, the prevalence of body dissatisfaction in adolescent girls is higher than in boys [5] and overweight adolescents are more body dissatisfied compared to adolescents of normal body weight [11,12]. Adolescent girls’ body dissatisfaction is associated with a higher drive for thinness, while boys with body dissatisfaction demonstrate a higher drive for muscularity [13]. Body dissatisfaction is associated with anomalous eating behaviours and studies demonstrate that unhealthy eating behaviours are a predictor of a higher risk of eating disorders [7]. Finally, some data exists which shows that body dissatisfaction is related to the development of obesity in later life and might be a precursor of eating disorders [14,15]. 

Studies have shown that adolescents’ body dissatisfaction and body image concerns are associated with the accuracy of the body weight perception [16]. Research in representative adolescent samples show that adolescents, especially girls, demonstrate an inadequate evaluation of their body weight with a high tendency to overestimate it [6]. Body weight overestimation is related to overweight and obesity development in later years [16]. Studies have shown that both over- and under-estimation are associated with the use of unhealthy weight control behaviours and a depressive mood [6]. Moreover, a study by Fan and Jin [17] demonstrated that, irrespective of weight status, adolescents who perceive themselves as overweight have a stronger intention to lose weight, but do not develop better eating and exercise habits, compared with their counterparts of the same gender and the same reported weight status [17]. However, a study in English adolescents demonstrated a relatively low prevalence of body weight overestimation, but almost half of the boys and a third of the girls with a too high body mass index (BMI) reported that their weight was normal, which is a big cause for concern [18]. Therefore, adequate body weight estimation is important for the development of healthy body image and a healthy lifestyle. 

Studies show that a higher BMI, body dissatisfaction and overestimation of body weight are associated with a restricted food intake, the use of laxatives and diuretics, purging, the use of diet pills, and induced vomiting [16,19]. Dieting together with unhealthy weight control behaviours may lead to eating disorders and obesity in later life. Therefore, the analysis of nutrition and nutritional habits is equally important in trying to understand the complexity of the development of obesity which is increasing worldwide. In their review, Spencer, Rehman, and Kirk [20] concluded that the majority of studies focused on adolescent girls’ dieting and unhealthy weight control behaviours, rather than on factors of nutrition [20,21]. For example, the study on a representative sample in Lithuania showed that skipping breakfast and a lower frequency of meals were more prevalent in adolescents with a higher BMI [22]. Furthermore, a study by Neumark–Sztainer et al. [16] demonstrated that skipping meals and not eating enough is related to a further increase in BMI, both in normal weight and overweight adolescents. A study by Lampard et al. [23] showed that exclusively healthy weight control behaviours were more prevalent in adolescent girls who were not overweight compared to overweight and obese girls. It was also found that adolescents who reported lower body fat, lower body dissatisfaction, and a higher self-esteem were more likely to report a healthier eating behaviour profile than those adolescents who reported high body fat, high body dissatisfaction, and low self-esteem [23]. A nationwide study by Hsu et al. [24] demonstrated that an underestimation of body weight in overweight Asian adolescents was associated with lower breakfast consumption, self-induced vomiting as a weight control strategy, fried food consumption, and engaging in vigorous physical activities [24]. 

While body mass control is related to physical activity and exercising, it is also important to explore the associations between physical activity and body image in adolescents. Studies have shown that the associations between body image concerns and physical activity are not consistent. Some studies have claimed that exercise helps to improve body image in adolescents [25,26,27]. However, other studies have shown that physical activity is not related to body image satisfaction in girls [28]. There is some evidence that body weight evaluation plays an important role in the relationship between body image and physical activity. A study by Gillison, Standage, and Skevington demonstrated that adolescents perceiving themselves as overweight and pressured into losing weight, endorsed extrinsic weight-related goals for exercise (i.e., exercise for weight loss). Extrinsic goals negatively predicted self-determined motivation and exercise behaviour. [3]. However, physical activity is often neglected in studies analysing body image, body weight evaluation, dieting and nutrition in adolescents. Thus, this study is attempting to fill this gap. 

The analysis of associations between adolescents’ body image, body weight evaluation, and lifestyle (nutrition and physical activity) is important for health promotion and health education programs. As the overweight and obesity numbers worldwide are increasing [29], significant efforts are devoted to healthy nutrition and physical activity promotion in schools’ health education. Studies have suggested that obesity prevention might be a complex issue involving behavioural lifestyle changes (healthy nutrition and physical activity) and developing healthy body image and self-esteem [30]. Thus, it is important to understand how those variables are interrelated. To our knowledge, there is a lack of studies examining the associations between body image, body weight perceptions and healthy nutrition related factors such as breakfast consumption, number of meals, consuming vegetables and fruit in adolescents, especially in boys. Moreover, there is a lack of such studies in the samples of Eastern Europe. 

Thus, the aim of the study was to explore the associations between body image concerns, disordered eating, self-esteem, nutrition habits and physical activity in the mixed sample of adolescents. In this study we expected that students with the higher BMI will demonstrate higher body dissatisfaction, drive for thinness, higher social physique anxiety and disordered eating, lower physical activity and self-esteem, and more unhealthy nutrition profile, controlling for gender. Furthermore, we hypothesised that adolescents overestimating their body weight will express more negative body image, higher disordered eating, lower physical activity and self-esteem, and more unhealthy nutrition pattern controlling for gender and BMI. 

## 2. Material and Methods

### 2.1. Study Participants

A total of 579 adolescents (299, 51.6% were girls) participated in the study. Participants were in the 9th grade of a randomly-selected public high schools in Kaunas (the second largest city of Lithuania). The participants ranged in age from 14–16 years old (*M* = 15.0, SD = 0.4).

### 2.2. Procedure

Respondents provided their answers filling in questionnaires consisting of a battery of self-report questionnaires designed to measure study variables. These results are part of a larger experimental study which was supported by the Lithuanian Council of Science (MIP—22/2010). The Lithuanian Committee of Bioethics approved the research (no. BE-2-62). 

### 2.3. Measures

Demographics. Participants were asked to indicate their age and date of birth. 

Body mass index was assessed using self-reported height and body weight. As recommended by the International Obesity Task Force (IOTF) cut-offs, the sample was classified into four body mass categories according to percentiles: below the 5th percentile was thin, between the 5th and 84th was normal weight, between the 85th and 94th was overweight and ≥ the 95th percentile was obese [31]. The prevalence of obesity was only 1.4%, thus, in the further analyses, the overweight and obesity categories were combined. The results showed that 12.7% of boys and 9.7% of girls were either overweight or obese.

Body Weight Discrepancy was measured as the difference between self-reported body weight and perceived ideal body weight. A negative value of body weight discrepancy indicated a desire to lose weight, zero satisfaction with their current weight, and a positive value—the desire to gain weight. The higher discrepancy showed a higher dissatisfaction with the current body weight.

Body weight perception was assessed as the discrepancy between current self-reported body weight and the reported desire to lose or to gain weight. Adolescents of a normal body weight, yet desiring to lose it, were attributed to the overestimation group. Next, adolescents were classified as adequately evaluating their body weight if their desire to lose or gain some weight was in concordance with their current body weight. Finally, overweight and obese adolescents who considered their weight as normal or even desired to gain some weight were attributed to the underestimation category.

Physical Activity was assessed using the leisure time exercise questionnaire (LTEQ) [32]. This instrument measures mild, moderate and strenuous physical activity over a week. The number of bouts of mild exercise was multiplied by 3, moderate exercise by 5 and strenuous exercise by 9, resulting in a final score of physical activity which provides a total metabolic equivalent by each intensity level. A higher score indicated a higher physical activity (PA) in each of the three levels. An additional question was used to assess the frequency of vigorous exercise (rarely/never, sometimes, or often). 

Nutrition was assessed by a food frequency questionnaire containing 11 groups of different foods and approved by the national survey of Health Behaviour in School Children [33]. An additional battery of questions was used to examine the regularity of meals, frequency of having breakfast, the choice of particular food groups and drinks for snacks and refreshments, and following a healthy diet regimen. The nutrition score consisted of nutrition habits. It summarised eating and breakfast frequency, eating between meals, regular eating time, choices of some foods and following a healthy diet regimen. A higher nutrition score indicated healthier nutrition habits.

Eating Disorder Inventory—3 (EDI-3) [34] was used to assess the attitudes and behaviours related to eating, weight and body image. In this study we used two subscales: Drive for Thinness (DT), and Body Dissatisfaction (BD). The DT scale assesses an extreme desire to be thinner, concerned with dieting and a preoccupation with weight and an intense fear of gaining weight. It consists of seven items with Likert-type answers from always (6) to never (0). Higher scores indicate a higher drive for thinness. The BD scale consists of 10 items and assesses discontentment with the overall shape and size of regions of the body that are of an extraordinary concern to those who have eating disorders. Higher scores indicate a higher body dissatisfaction. The scales had adequate psychometric qualities in adolescent non-clinical samples. In our sample, the internal consistency of the drive for thinness and body dissatisfaction subscales was: Cronbach α = 0.86 and 0.88. 

Eating Attitudes Test—26 (EAT-26) [35] was used to measure the participants’ disordered eating and risk for eating disorders. EAT-26 consists of 26-items scored on a Likert scale (0—never to 6—always) and the items are summed to obtain an overall score (range = 0 to 78). Individuals with an overall score of 20 or higher are considered to be at risk of an eating disorder. EAT—26 consists of three subscales (bulimia, food preoccupation, and oral control). In this study we used the scale of 26 items with a higher score indicating a greater risk for eating disorders. Internal consistency of the scale was good, Cronbach α = 0.88. Further analyses were performed in three groups: ≤10 scores, 10–20 scores, and ≥20 scores.

A nine-item Social Physique Anxiety Scale (SPAS) [36] was used to measure the degree of anxiety that an individual’s experience when presenting the physique in an evaluative context. Participants responded to items on a five-point Likert scale from 1 = “not at all true for me” to 5 = “very true for me”). A higher score shows a higher social physique anxiety. For the present study, the internal consistency of the scale was good (Cronbach α = 0.88).

Muscularity—Oriented Body Image Attitudes were assessed using the Drive for Muscularity Scale (DMS) [13]. This scale examines a person’s attitudes and behaviours reflecting a preoccupation with muscularity. The scale consists of 15 items rated a on 6-point scale ranging from 1 (never) to 6 (always). Higher scores indicate a higher drive for muscularity. The Muscularity—Oriented Body Image Attitudes subscale consists of seven items assessing a desire to have a more muscular body, i.e., “I wish that I was more muscular”, or “I think I would be more confident if I had more muscle mass”, etc., Higher scores indicate a higher intention to have a muscular—oriented body image. For the present study, the internal consistency of the scale was good (Cronbach α = 0.84). This subscale was only provided for boys. 

The Rosenbergs’ Self-Esteem Scale [37] is the most widely used measure of global self-esteem and has been determined to be valid and reliable among students. The scale consists of 10 items rated on a four-point Likert scale (from 1 = “not at all true for me” to 4 = “very true for me”) yielding scores from 10 to 40. Higher scores reflect a greater level of self-esteem. For the present study, the internal consistency of the scale was good (Cronbach α = 0.82). 

### 2.4. Statistical Analysis

First, descriptive statistics and distribution normality testing of the continuous variables were performed, the results were presented as means ± standard deviations and as percentages according to the type of variable. The relationships between the categorical variables were analysed using the χ2 test. The Kolmogorov-Smirnov test was used to determine the normality distribution of the variables. For normally distributed continuous variables comparisons between boys and girls were made using an independent sample *t*-test, for non-normally distributed variables the Mann-Whitney U test was used. Comparisons of ordered nutrition habits in body weight and disordered eating groups were performed by the Kruskal–Wallis test. Finally, an analysis of covariance was employed to test the hypotheses about the differences of body image concerns in the groups of BMI and body weight estimation controlling for gender. Statistical analyses were conducted using IBM SPSS Statistics 25 (IBM Corp., Armonk, NY, USA).

## 3. Results

The descriptive statistics showed that girls’ BMIs were lower than boys (Table 1). As expected, girls overestimated body weight more frequently than boys, while more boys demonstrated body weight underestimation. However, girls more often demonstrated adequate body weight estimation as compared to boys. The analysis of the differences between desired and current body mass showed that boys more frequently wanted to increase their body weight, while girls demonstrated the opposite. Furthermore, as expected, boys were more physically active in leisure time and vigorous physical activity in boys was more prevalent when compared to girls. No significant differences were observed in disordered eating between genders. The analysis of the nutrition score revealed no gender differences, however, boys reported a higher number of meals and more frequent breakfast consumption compared to girls. As expected, boys demonstrated higher self-esteem, lower body dissatisfaction, lower social physique anxiety, and a lower drive for thinness compared to girls. The drive for muscularity analysis showed a similar average score as in other Western samples [38]. 

Next, we analysed the associations between body image concerns, anomalous eating, nutrition, self-esteem and leisure time exercise in BMI groups (Table 2). The associations between study variables were in the expected direction. The overweight group demonstrated the highest body image concerns, the highest score in disordered eating, the lowest self-esteem, yet no significant differences were observed in the drive for muscularity, leisure time physical activity, and nutrition scores. 

Further, we explored the nutrition habits in the BMI groups of boys and girls. In boys, we found no significant differences in the nutrition habits in normal weight, underweight, and overweight groups with the exception of the lowest consumption of fruits and berries in overweight boys as compared to boys of a normal weight and underweight (Table 3).

Furthermore, we found that overweight girls consumed less fats, spreads, and oils compared to girls of a normal weight and underweight and more frequently skipped breakfast (Table 4).

Next, we analysed body image concerns and nutrition in the groups for the accuracy of current body weight estimation controlling for gender and BMI (Table 5). As expected, the results showed that the body weight overestimation group suffered from the highest body image concerns and social physique anxiety, anomalous eating and lowest self-esteem. However, there were no differences in the drive for muscularity between those groups. Further, we observed no differences in leisure time physical activity and nutrition scores controlling for gender and BMI. 

Furthermore, we tested the correlations between BMI, body image concerns, anomalous eating, nutrition, and physical activity in boys and girls (tables are not presented). The analysis demonstrated the associations of the expected direction between body image variables and disordered eating in both genders, while the correlations were stronger in girls. Correlations between disordered eating and body dissatisfaction, drive for thinness, and social physique anxiety were as follows: 0.44, 0.53, 0.46 in girls and 0.16, 0.35, and 0.32 in boys, *p* < 0.01. As expected, muscular orientated body image attitudes in boys were negatively correlated with body dissatisfaction, drive for thinness, and social physique anxiety (−0.13, −0.14, and −0.38, *p* < 0.05) but positively correlated with self-esteem (0.25, *p* < 0.01). Self-esteem was negatively associated with body image concerns in both genders. Correlations between self-esteem and body dissatisfaction, drive for thinness and social physique anxiety were as follows: −0.49, −0.31, −0.56 in girls and −0.39, −0.19, and −0.50 in boys, *p* < 0.01. Further correlation analyses demonstrated that disordered eating was not correlated with nutrition habits in boys, however, higher disordered eating scores in girls were associated with a lower breakfast frequency (−0.23, *p* < 0.01), a lower number of meals (−0.13, *p* < 0.05), and a lower consumption of fish (−0.13, *p* < 0.05). In addition, muscular orientated body image attitudes were not associated with nutrition habits in boys, yet, boys’ body dissatisfaction and social physique anxiety were related to a lower number of meals (accordingly −0.22 and −0.14, *p* < 0.05). In girls, body dissatisfaction, drive for thinness, and social physique anxiety were negatively correlated with the number of meals (accordingly −0.25, −0.20, and −0.19, *p* < 0.01), while the drive for thinness was negatively related to breakfast consumption (−0.14, *p* < 0.05).

Next, we explored the nutrition habits between the body mass perception groups in boys and girls. The results in boys demonstrated that body weight underestimation in boys was related to a higher consumption of milk, cheese, yogurt, fat, spreads, and oils, a higher number of meals per day, and a higher frequency of having breakfast (Table 6).

The nutrition habits in girls were similar between the body weight estimation groups, however, it was found only one significant difference in eating more sweets (cakes, candies, and chocolate) in underestimation group as compared to the groups of adequate body weight estimation and overestimation, mean ± SD accordingly: 2.39 ± 0.83; 2.04 ± 0.82; 1.90 ± 0.90, *p* = 0.018 (table is not presented). 

Finally, we analysed the nutrition habits in disordered eating groups. In boys, the group with the highest disordered eating consumed more fish when compared to the middle and lowest score groups (Table 7). Furthermore, this group demonstrated the lowest consumption of eggs and the lowest frequency of having breakfast.

In girls, the group with the highest disordered eating consumed less meat as compared to the middle and lowest scoring groups (Table 8). Next, this group reported they consumed less fats, spreads, and oils, and the lowest breakfast consumption.

## 4. Discussion

The aim of the present study was to explore the associations between body image, disordered eating, nutrition habits, self-esteem, and physical activity in the mixed sample of adolescents. We expected that students with a higher BMI will demonstrate higher body dissatisfaction, a drive for thinness, higher social physique anxiety, disordered eating, lower physical activity and self-esteem, and an unhealthier nutrition pattern controlling by gender. The results partially confirmed our first assumption. As expected, we found negative associations between a higher BMI and body image concerns, social physique anxiety, self-esteem, and disordered eating. These results are in line with the findings of other studies reporting a lower psychosocial well-being of overweight adolescents of both genders [2,7,11,12,39,40]. 

Yet, we found no associations between BMI, nutrition score, and leisure time exercising. It shows that despite body weight dissatisfaction and body image concerns, overweight or obese adolescents do not use physical activity or healthy eating, which means helping them to reduce body weight. Instead, overweight boys eat less fruits and berries and girls report less frequent breakfast consumption. This finding is in line with other representative studies in Lithuania [22] and studies demonstrating that adolescents attempting to lose weight do not increase the consumption of vegetables and fruit [39]. The study of Bibiloni et al. [11] also showed that overweight and obese adolescents who wanted to have a thinner body reported lower eating numbers and skipping breakfast (girls). It emphasises the importance of health education, teaching adolescents and their family’s healthy nutrition and physical activity principles when reducing body weight. 

Body dissatisfaction in both genders and a drive for thinness and social physique anxiety in girls correlated with a lower number of meals in the present study. These findings go in line with other studies. A study by Cassia Ribeiro-Silva et al. [40] showed that overweight and obese adolescents with a higher body dissatisfaction demonstrated so called “restrictive pattern” of eating. As expected, disordered eating in our study was associated with an unhealthier eating pattern, particularly eating less fish and eggs in boys, less meat in girls and a lower breakfast consumption in both genders. Thus, our results emphasise the necessity to help overweight and obese adolescents to cope with body image concerns and to develop healthy body image and healthy eating habits. 

However, the surprising finding was that overweight and obesity was associated with having fewer sweets (cakes, candies, chocolate), consuming less fats, spreads, and oils in girls. This study showed that adolescents with the higher BMI demonstrated higher body dissatisfaction, thus, these findings go in line with other studies demonstrating that overweight adolescents with higher body image dissatisfaction eat fewer fats and sugar-containing snack foods [11,40,41]. It seems that adolescent girls try to avoid sweets and fats, yet they lack healthy nutrition and physical activity habits. These findings might also be explained as the social desirability of responses. Studies show that overweight adolescents tend to underreport their energy intake [42]. However, in their longitudinal study Hume, Yokum, and Stice demonstrated that low energy flux (low calories consumption and low energy expenditure) but not energy surfeit (high calorie consumption and low energy expenditure) are associated with future increases in body fat in adolescents and young women [43]. Thus, it might also be that adolescents desiring to lose weight are episodically or constantly dieting and then have episodes of binge eating, yet we cannot draw this conclusion from the present study. Future research might benefit from assessing calorie intake in similar studies. 

Further, we hypothesised that adolescents overestimating their body weight will express a more negative body image, disordered eating, lower physical activity, lower self-esteem, and an unhealthier nutrition profile controlling for gender and BMI. We found that body weight overestimation was more prevalent in girls, yet underestimation was more common for boys. These findings coincide with other studies showing that girls want to decrease body weight while boys want to increase muscle mass [11]. The second assumption was partially confirmed as well. As expected, we found that adolescents overestimating their weight demonstrate higher body dissatisfaction, a drive for thinness, social physique anxiety, higher disordered eating, and lower self-esteem. These findings are in line with other studies [7,11,12,39,40]. It adds to the knowledge that being overweight or obese and believing that their weight is too high is equally detrimental for adolescents’ body image and psycho-social health. 

Further analyses demonstrated that the associations between body weight overestimation and nutrition depends on gender. Boys, overestimating their body weight, ate less milk, cheese, yogurt, fats, spreads, and oils, reported a lower number of meals per day, and a lower frequency of having breakfast. On the other hand, weight underestimation in boys was associated with a higher consumption of milk products, fats, spreads, oils, and a higher number of meals. This is in line with other studies reporting that boys’ body image is based on a drive for muscularity and muscle—body building eating behaviour (keeping a high protein diet) [11]. This nutrition pattern might also lead to the development of overweight in normal weight adolescents underestimating their weight which is detrimental for those who are already overweight or obese, but they do not realise this. In conclusion, it seems that, for boys, overestimation and underestimation of body weight are equally problematic. 

Our study demonstrated that girls with a higher BMI and those who overestimate their body weight and want to decrease it, demonstrate higher body image concerns, social physique anxiety, disordered eating and lower self-esteem. Overweight and obese girls use fewer spreads and oils, but report having breakfast less frequently when compared to those with normal body weight. The girls underestimating their body weight demonstrated less body image concerns and higher self-esteem, yet, they more frequently consumed sweets. Our study showed no significant differences in physical activity in the BMI and body weight estimation groups. Thus, being overweight or obese, and/or believing that body weight is too high is an equally detrimental situation for girls’ body image and self-esteem. This study confirms that overweight and obese female adolescents, or those overestimating body weight and seeking to decrease it, demonstrate no better nutrition or leisure time physical activity habits than adolescents with a normal weight.

Finally, we found no associations between body image concerns, body weight perception and leisure time physical activity in adolescents of both genders. These results are in line with other studies [2,27,28]. However, the interpretation of this finding is limited as we did not assess the nature and/or motivation of participation in exercise activity. Future studies might address this issue. 

The study expands the existing literature on the associations between body image concerns and the lifestyle of adolescents [2,6,9,11,17]. The findings of the present study have important implications for public health, health promotion and education. Firstly, this knowledge adds to the understanding that we should reframe the direction of policy and practice when implementing informing public health campaigns which try to shame larger-bodied people and to try to get them to exercise or to eat healthy. The principle “first do no harm” must be followed. A study by Marks (2015) demonstrated that the development of overweight and obesity is a complex issue depending on circumstances weakly controlled by human efforts (i.e., socioeconomic status) [15]. Therefore, we should avoid the “victim blaming” position in public health. Instead, it is important to support developing a positive body image in adolescents of all body sizes and to provide them evidence-based health education which teaches them the principles of healthy nutrition and exercise related weight control. Studies show that weight-related problems continue from adolescence to adulthood [44] and dissatisfaction with body weight may lead to eating disorders and overweight or obesity in future life [16,41]. Thus, our study also supports the idea that the prevention of eating disorders and obesity must be integrated [45]. 

## 5. Conclusions

Adolescents with a higher BMI and body weight overestimation demonstrated higher body image concerns, lower self-esteem, and a poorer eating-related behavioural profile. Body image concerns and body weight overestimation do not promote healthy behaviour in adolescents. It is critical to promote positive body image, adequate body weight evaluation, self-esteem, and healthy lifestyle in health promotion and health education programs for adolescents of both genders and different BMIs. These results support the idea that teaching about positive body image and adequate body weight estimation must be a part of all evidence-based healthy nutrition, physical activity, and weight control-related health education programs. 

### Study Strengths and Limitations

An important strength of this study relatively large and random sample size, the use of internationally sound measures to assess study variables, and the inclusion of the variable assessing factual nutrition in adolescents. By determining associations between BMI, body weight estimation, body image concerns, physical activity, and nutrition this study may contribute to a shift of attention from body weight as the target of health education, towards the promotion of the healthy body image and lifestyle. 

Despite these strengths, this research has some limitations. First, this study cannot infer causality because of the cross-sectional nature of the research. The second limitation is the self-reported data of body height and weight. As the prevalence of overweight and obesity in Lithuanian adolescents is low [22], we found relatively small numbers of overweight or obese adolescents in the present sample. Thus, it was not possible to analyse in depth the effects of body image and body dissatisfaction on the nutrition habits. Further studies in Western samples with a higher overweight or obesity prevalence should address this question. Finally, there is a risk that adolescents who wish to be thinner might overestimate healthy food and underestimate unhealthy food consumption [11]. As it is one of the first studies in this topic in Eastern Europe, future research should test our findings. 

## Figures and Tables

**Table 1 ijerph-16-00864-t001:** Descriptive statistics for the sample (*n* = 579).

Characteristics		Boys (*n* = 280)	Girls (*n* = 299)	*p*
Age, years (m ± SD)		15.0 ± 0.4	14.9 ± 0.4	0.011
Body mass, %	Underweight	7.2	15.2	0.008
	Normal weight	80.1	75.1	
	Overweight/obesity	12.7	9.7	
Body mass perception adequacy, %	Overestimation	13.2	55.4	<0.0001
	Adequate estimation	25.6	34.4	
	Underestimation	61.2	10.2	
Vigorous exercising	Never or rarely	6.3	25.9	<0.0001
	Sometimes	38.1	57.8	
	Often	55.6	16.3	
Disordered eating	<10 scores, %	84.3	76.9	0.052
	10–19 scores, %	8.6	14.7	
	≥20 scores, %	7.1	8.4	
Difference between desired and estimated body mass, m ± SD		7.5 ± 12.0	−2.9 ± 5.8	<0.0001
Leisure time exercise, m ± SD		66.7 ± 34.5	47.8 ± 27.2	<0.0001
Nutrition score, m ± SD		19.1 ± 4.0	18.6 ± 3.6	0.200
No. of meals per day, m ± SD		4.1 ± 1.4	3.7 ± 1.2	<0.0001
Frequency of having breakfast *, m ± SD		2.3 ± 0.9	2.0 ± 1.0	0.003
Self-esteem, m ± SD		30.9 ± 4.3	29.1 ± 4.7	<0.0001
Body dissatisfaction, m ± SD		1.2 ± 1.0	1.7 ± 1.2	<0.0001
Drive for thinness, m ± SD		0.6 ± 0.9	1.0 ± 1.1	<0.0001
Muscularity—oriented body image attitudes, m ± SD		3.68 ± 1.21	-	-
Social physique anxiety, m ± SD		2.1 ± 0.8	2.6 ± 0.9	<0.0001

* from 0 (never) up to 3 (always). *n* = number, % = percentage, m = mean, SD = standard deviation, *p* = statistical significance level.

**Table 2 ijerph-16-00864-t002:** Comparison of body image concerns and exercise between body mass groups (m, 95% CI) ^1^, (*n* = 579).

Variables	Underweight	Normal Body Mass	Overweight	*p*
Body dissatisfaction	1.04 (0.78–1.32)	1.41 (1.30–1.51)	2.23 (1.95–2.50)	<0.0001
Drive for thinness	0.28 (0.04–0.52)	0.85 (0.75–0.94)	1.27 (1.02–1.51)	<0.0001
Muscularity—oriented body image attitudes (boys)	3.26 (2.71–3.81)	3.66 (3.49–3.82)	3.86 (3.44–4.28)	0.226
Social physique anxiety	2.03 (1.82–2.23)	2.29 (2.21–2.37)	2.88 (2.67–3.10)	<0.0001
Disordered eating	4.48 (2.26–6.70)	6.35 (5.51–7.19)	8.27 (6.04–10-49)	0.062
Self-esteem	30.1 (28.9–31.2)	30.3 (29.8–30.7)	28.1 (26.9–29.2)	0.003
Body weight discrepancy	−8.97 (−11.11–(−6.82))	−2.53 (−3.34–(−1.73))	8.37 (6.23–10.51)	<0.0001
Leisure time exercise score	55.4 (46.7–64.2)	57.2 (54.0–60.5)	58.6 (49.6–67.5)	0.883
Nutrition score	19.5 (18.6–20.5)	18.7 (18.3–19.0)	18.8 (17.9–19.8)	0.250

^1^ controlled for gender (analysis of covariance). m = mean, 95% CI = 95% confidence interval, *p* = statistical significance level.

**Table 3 ijerph-16-00864-t003:** Comparison of nutrition habits between body mass groups in boys (m ± SD) ^1^, (*n* = 579).

Foods and Drinks	Underweight	Normal Weight	Overweight, Obesity	*p*
Cereals, breads, potatoes, rise, pasta	2.28 ± 0.83	2.23 ± 0.90	2.37 ± 0.81	0.810
**Fruits and berries**	**2.26 ± 0.93**	**2.05 ± 0.84**	**1.68 ± 0.79**	**0.027**
Vegetables	2.25 ± 0.86	2.07 ± 0.85	2.03 ± 0.77	0.643
Meat	2.44 ± 0.86	2.40 ± 0.79	2.23 ± 0.85	0.468
Fish	0.79 ± 0.42	1.16 ± 0.79	0.90 ± 0.60	0.057
Milk, cheese, yogurt	2.28 ± 0.83	2.26 ± 0.91	2.10 ± 0.91	0.560
Eggs	1.39 ± 0.98	1.43 ± 0.85	1.16 ± 0.78	0.298
Legumes	0.84 ± 0.77	1.15 ± 0.84	1.06 ± 0.81	0.245
Sweets (cakes, candies, chocolate)	2.05 ± 0.78	1.85 ± 0.90	1.52 ± 0.72	0.054
Fats, spreads and oils	2.05 ± 0.91	1.96 ± 0.95	1.84 ± 0.82	0.601
Soft drinks with sugar	1.95 ± 1.08	1.51 ± 0.95	1.37 ± 0.89	0.119
No. of meals per day	4.11 ± 0.99	4.19 ± 1.40	3.84 ± 1.69	0.093
Frequency of having breakfast	2.37 ± 0.90	2.29 ± 0.92	2.12 ± 1.05	0.646

^1^ higher score indicate more frequent consumption of particular food groups or number of meals. m = mean, SD = standard deviation, *p* = statistical significance level. Statistically significant differences are in bold text.

**Table 4 ijerph-16-00864-t004:** Comparison of nutrition habits between body mass groups in girls (m ± SD) ^1^, (*n* = 579).

Foods and drinks	Underweight	Normal Weight	Overweight, Obesity	*p*
Cereals, breads, potatoes, rise, pasta	2.20 ± 0.88	2.19 ± 0.92	2.32 ± 0.82	0.847
Fruits and berries	1.93 ± 0.87	2.07 ± 0.85	2.07 ± 0.81	0.588
Vegetables	2.23 ± 0.81	2.19 ± 0.82	2.21 ± 0.88	0.956
Meat	2.41 ± 0.84	2.24 ± 0.87	2.18 ± 0.82	0.330
Fish	0.98 ± 0.83	1.09 ± 0.77	1.11 ± 0.69	0.518
Milk, cheese, yogurt	2.28 ± 0.88	2.26 ± 0.84	2.21 ± 0.88	0.929
Eggs	1.33 ± 0.75	1.21 ± 0.79	1.18 ± 0.67	0.654
Legumes	1.14 ± 0.89	0.91 ± 0.79	0.89 ± 0.83	0.302
**Sweets (cakes, candies, chocolate)**	**2.33 ± 0.81**	**1.98 ± 0.89**	**1.71 ± 0.66**	**0.01**
**Fats, spreads and oils**	**2.33 ± 0.84**	**1.90 ± 1.02**	**1.86 ± 0.85**	**0.031**
Soft drinks with sugar	1.47 ± 1.08	1.23 ± 0.98	0.93 ± 0.94	0.091
No. of meals per day	3.86 ± 0.93	3.68 ± 1.24	3.25 ± 0.89	0.057
**Frequency of having breakfast**	**2.32 ± 0.88**	**2.01 ± 1.01**	**1.75 ± 0.97**	**0.046**

^1^ higher score indicate more frequent consumption of particular food groups or number of meals. m = mean, SD = standard deviation, *p* = statistical significance level. Statistically significant differences are in bold text.

**Table 5 ijerph-16-00864-t005:** Comparison of body image concerns and exercise between body weight discrepancy groups (m, 95% CI) ^1^, (*n* = 579).

Variables	Overestimation	Adequate Estimation	Underestimation	*p*
Body dissatisfaction	1.97 (1.82–2.13)	1.36 (1.20–1.52)	1.05 (0.88–1.21)	<0.0001
Drive for thinness	1.33 (1.19–1.47)	0.69 (0.55–0.83)	0.44 (0.29–0.59)	<0.0001
Muscularity—oriented body image attitudes (boys)	3.59 (3.16–4.01)	3.71 (3.41–4.01)	3.63 (3.43–3.82)	0.875
Social physique anxiety	2.67 (2.55–2.80)	2.27 (2.15–2.39)	2.03 (1.89–2.16)	<0.0001
Disordered eating	8.23 (6.92–9.54)	5.78 (4.46–7.09)	4.68 (3.29–6.06)	0.002
Self-esteem	29.3 (28.6–30.0)	30.0 (29.3–30.7)	30.7 (30.0–31.5)	0.046
Body weight discrepancy	−3.23 (−4.27–2.20)	0.20 (−0.84–1.23)	9.07 (7.97–10.16)	<0.0001
Leisure time exercise score	55.5 (50.1–61.0)	55.3 (49.9–60.7)	60.5 (55.0–65.9)	0.384
Nutrition score	18.7 (18.1–19.3)	19.3 (18.7–19.9)	18.2 (17.6–18.9)	0.055

^1^ controlled for gender and body mass index (analysis of covariance). m = mean, 95% CI = 95% confidence interval, *p* = statistical significance level. Statistically significant differences are in bold text.

**Table 6 ijerph-16-00864-t006:** Comparison of nutrition habits between body mass perception groups in boys (m ± SD) ^1^, (*n* = 579).

Foods and Drinks	Overestimation	Adequate Estimation	Underestimation	*p*
Cereals, breads, potatoes, rise, pasta	2.03 ± 0.93	2.29 ± 0.86	2.32 ± 0.85	0.250
Fruits and berries	1.97 ± 0.86	1.94 ± 0.87	2.09 ± 0.84	0.426
Vegetables	2.19 ± 0.79	2.22 ± 0.77	2.01 ± 0.87	0.247
Meat	2.13 ± 0.87	2.35 ± 0.84	2.46 ± 0.73	0.109
Fish	1.26 ± 0.82	1.03 ± 0.75	1.10 ± 0.75	0.420
**Milk, cheese, yogurt**	**1.78 ± 1.04**	**2.30 ± 0.88**	**2.32 ± 0.84**	**0.016**
Eggs	1.28 ± 0.88	1.31 ± 0.83	1.46 ± 0.86	0.390
Legumes	1.03 ± 0.88	1.10 ± 0.80	1.15 ± 0.85	0.684
Sweets (cakes, candies, chocolate)	1.84 ± 0.88	1.73 ± 0.85	1.87 ± 0.90	0.509
**Fats, spreads and oils**	**1.47 ± 1.05**	**1.95 ± 0.93**	**2.05 ± 0.88**	**0.013**
Soft drinks with sugar	1.59 ± 1.04	1.59 ± 0.97	1.48 ± 0.92	0.789
**No. of meals per day**	**3.82 ± 1.90**	**3.78 ± 0.98**	**4.25 ± 1.13**	**0.002**
**Frequency of having breakfast**	**1.68 ± 1.17**	**2.35 ± 0.94**	**2.37 ± 0.84**	**0.003**

^1^ higher scores indicate more frequent consumption of particular food group or number of meals. m = mean, SD = standard deviation, *p* = statistical significance level. Statistically significant differences are in bold text.

**Table 7 ijerph-16-00864-t007:** Comparison of nutrition habits of adolescent boys in groups of disordered eating (m ± SD) ^1^, (*n* = 579).

Foods and Drinks	Disordered Eating			*p*
	<10 Scores	10–19 Scores	≥20 Scores	
Cereals, breads, potatoes, rise, pasta	2.26 ± 0.88	2.14 ± 0.85	2.40 ± 0.99	0.533
Fruits and berries	2.03 ± 0.86	2.14 ± 0.79	1.87 ± 0.83	0.631
Vegetables	2.06 ± 0.84	2.27 ± 0.88	2.27 ± 0.80	0.306
Meat	2.36 ± 0.81	2.38 ± 0.87	2.47 ± 0.64	0.950
**Fish**	**1.04 ± 0.71**	**1.36 ± 0.95**	**1.50 ± 1.02**	**0.044**
Milk, cheese, yogurt	2.22 ± 0.89	2.48 ± 0.81	2.20 ± 1.08	0.416
**Eggs**	**1.43 ± 0.83**	**1.45 ± 1.01**	**0.71 ± 0.61**	**0.007**
Legumes	1.10 ± 0.82	1.14 ± 0.91	1.38 ± 1.12	0.578
Sweets (cakes, candies, chocolate)	1.85 ± 0.86	1.73 ± 0.99	1.64 ± 1.08	0.692
Fats, spreads and oils	1.99 ± 0.92	1.71 ± 1.10	1.67 ± 1.11	0.308
Soft drinks with sugar	1.43 ± 0.95	1.86 ± 1.04	1.93 ± 0.96	0.024
No. of meals per day	4.11 ± 1.31	4.22 ± 1.28	4.44 ± 2.45	0.825
**Frequency of having breakfast**	**2.34 ± 0.88**	**2.35 ± 1.03**	**1.50 ± 1.15**	**0.002**

^1^ higher scores indicate more frequent consumption of particular food group or number of meals. m = mean, SD = standard deviation, *p* = statistical significance level. Statistically significant differences are in bold text.

**Table 8 ijerph-16-00864-t008:** Comparison of nutrition habits of adolescent girls in groups of disordered eating (m ± SD) ^1^, (*n* = 579).

Foods and Drinks	Disordered Eating			*p*
	<10 Scores	10–19 Scores	≥20 Scores	
Cereals, breads, potatoes, rise, pasta	2.19 ± 0.89	2.40 ± 0.85	2.13 ± 0.99	0.312
Fruits and berries	2.01 ± 0.83	2.00 ± 0.82	2.29 ± 1.00	0.221
Vegetables	2.17 ± 0.81	2.21 ± 0.87	2.36 ± 0.86	0.441
**Meat**	**2.29 ± 0.82**	**2.44 ± 0.91**	**1.76 ± 1.01**	**0.006**
Fish	1.07 ± 0.75	1.00 ± 0.82	1.20 ± 0.82	0.573
Milk, cheese, yogurt	2.22 ± 0.82	2.48 ± 0.89	2.13 ± 0.95	0.069
Eggs	1.24 ± 0.74	1.21 ± 0.86	1.04 ± 0.83	0.377
Legumes	0.94 ± 0.78	0.81 ± 0.82	1.08 ± 1.06	0.521
Sweets (cakes, candies, chocolate)	2.02 ± 0.85	2.02 ± 0.91	1.52 ± 0.98	0.062
**Fats, spreads and oils**	**2.01 ± 0.94**	**2.05 ± 1.07**	**1.35 ± 1.07**	**0.013**
Soft drinks with sugar	1.20 ± 0.96	1.47 ± 1.07	1.13 ± 1.15	0.261
No. of meals per day	3.69 ± 1.01	3.67 ± 1.27	3.48 ± 2.04	0.110
**Frequency of having breakfast**	**2.14 ± 0.94**	**1.86 ± 1.06**	**1.40 ± 1.23**	**0.004**

^1^ higher scores indicate more frequent consumption of particular food group or number of meals. m = mean, SD = standard deviation, *p* = statistical significance level. Statistically significant differences are in bold text.

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
