# Peer review of "Body Image Concerns and Body Weight Overestimation Do Not Promote Healthy Behaviour: Evidence from Adolescents in Lithuania"

_ijerph, 2019, doi:10.3390/ijerph16050864_

Reviewer 1 Report

You must continue in this line of  research and expand it

Author Response

Dear Reviewer,

Thank you very much for your remarks.

Reviewer 2 Report

The numerous issues in the abstract (some mentioned below) suggests that this paper is not ready to be considered for publication.

There are grammatical errors in the abstract alone. Mixed adolescents is not defined. There is no mention of the statistical methods used (or correction for the many tests that were done) so it is not possible to evaluate whether the results are presented appropriately. BMI%ile or BMI z-score should have been used instead of BMI. The conclusion is overstated as this is not a longitudinal study.

Author Response

Dear Reviewer,

Thank you very much for your remarks. Please find our comments attached.

Reviewer 3 Report

This is such important research- it is critical that health promotion, public health, and health education professionals understand that body dissatisfaction does not motivate healthy behaviours. I have been searching for articles that prove this for some time, and here it is!

I encourage you to use a title that makes this point – when I try to search for articles that present this sort of research I don’t often use many of the keywords in your title. Why not use something adapted from this sentence from your abstract “Body image concerns and body weight overestimation do not promote healthy behaviour: Evidence from adolescents in Lithuania"

I have a few suggestions:

-       In the abstract you mention that the overestimation leads to weight gain but you don’t have the data to support this in this cross-sectional study. You could make this point in the discussion with support from some of the US longitudinal research, e.g., Neumark-Stainer et al., 2006.

-       I think you could make a stronger case for the necessity of this particular research- saying that the obesity rates are rising is not helpful or a strong case for this research. You mention a number of studies that found similar findings to you in the discussion- what does your paper find that theirs does not?

-       The columns on the left of tables should be left justified as this makes it easier to read

-       In the discussion, you suggest that the high consumption of milk etc, and fats and spreads might lead to the development of overweight in boys… I agree that boys’ body image is complex, but I think that there needs to be some consideration of the complexity in terms of boys aiming to increase muscle, as opposed to increasing weight. In my research, boys report consuming more dairy, eggs, and meat in order to gain muscle…

-       You need to discuss the implications of this research- I think you should make the case that this knowledge should reframe the direction of policy and praxtice- e.g., informing public health campaigns to avoid shaming larger bodied people in order to try to get them to exercise or eat well, inform schools and GP’s that they should not be sending ‘BMI report cards’ home, etc.

-       Please consider carefully the important contribution that this research could make to the field, and to current policy and practice and represent these thoughts with a strong conclusion.

-       I think that you need to make the distinction between what you have found, and what this could lead to based on what others have found, in the sentence in the conclusion, line 386: “Summing it all, these findings coincide with other findings stating that body image concerns and body weight overestimation do not foster adolescents’ healthy behavior (healthy eating and physical activity) and may lead to eating disorders and overweight or obesity in future life [17; 39 389 ;40]. “

There are a few grammatical errors and typos throughout the text- please double check the entire manuscript. Some examples:

-       Page 9, line 312 ‘Study of Cassia …’ Shoul be ‘The study of ….’

-       Page 10, Line 350: Remove ‘situations’ from this sentence to improve grammar: “Thus, being overweight or obese, and/or thinking that body weight is too high is equally detrimental situations for girls’ body image and self-esteem”

I look forward to this paper being published so that I can write some media articles for the general public, fitness trainers, dieticians, public health professionals and physical education teachers in order to convince them that we don’t need to make young people feel bad about themselves in order for them to ‘be healthy’. I hope that you will do the same in your country and the broader European context.

Neumark-Sztainer, D., et al. (2006). "Does body satisfaction matter? Five-year longitudinal associations between body satisfaction and health behaviors in adolescent females and males." Journal of Adolescent Health 39: 244-251.

Author Response

Dear Reviewer,

Thank you very much for your remarks and valuable comments to improve our paper. All your remarks were accepted. Please find our comments attached.

Reviewer 4 Report

The authors are to be commended on this interesting study of adolescents and their BMI, exercise, nutrition and body image. The inclusion of a food frequency questionnaire along with measures of eating disorder behaviours, disordered eating, body image and physical activity data make this a comprehensive study. This is valuable research, and with the advice of a copy-editor experienced in the English language, this paper would be improved, as there are some sections that are difficult to follow or are grammatically incorrect.

I agree with the conclusions drawn by this study, and note that these may be strengthened to include a health at every size message: It is important to promote positive body image not only in both genders, but also in adolescents of every size or BMI.

Specific comments

The abstract is difficult to follow, and there are a few sentences that need to be revised to ensure that they make sense. The use of so many abbreviations makes it hard to read. While  I understand that adhering to a word limit might encourage the use of abbreviations, it might be worth limiting the use of these here to increase readability.

Almost half of the references given are older than 2015. This paper may benefit from some more recent research.

I would question the use of the term 'late adolescence' in relation to this sample.The authors state that this study is to fill the gap in research with older adolescents, such as those aged between 16 and 19 years of age, but the study sample is aged 14 to 16 years of age (Line 73 -74). Is this more likely to be termed 'middle adolescence'?

Could the authors explain the relationship between body weight and exercise further?  (Lines 84-89) This is important to explain the justification and context for this research.

What tools were used to measure Body Weight Discrepancy (BWD) and Body weight perception? (Lines 126 & 129)

The reference used in Line 204 is almost 20 years old.Is there a newer, more suitable reference available ?

MOBI is used as an abbreviation (line 253) without explaining it previously in the text.

In the tables in the results section, please give sample sizes for each table.

In Table 8 (line 293) the authors use the term 'eating disorders'. I am assuming they mean 'disordered eating' here rather than participants with a diagnosed eating disorder? Please clarify.

In the discussion (lines 317 to 324) it would be interesting to further explore the seeming paradox of the overweight girls eating less of the high calorie foods. Is this to do with self-report? Or social desirability of responses? These are touched on briefly in terms of reporting of weight but less so in terms of diet.

Author Response

(The authors gave the same response as above.)
